# Real-World Survival Impact of New Treatment Strategies for Lung Cancer: A 2000–2020 French Cohort

**DOI:** 10.3390/cancers16152768

**Published:** 2024-08-05

**Authors:** Clemence Basse, Matthieu Carton, Maud Milder, Romain Geiss, Pauline Du Rusquec, Catherine Daniel, Marie-Ange Massiani, Alain Livartowski, Nicolas Girard

**Affiliations:** 1Thoracic Oncology, Hôpital Institut Curie, St Cloud, 75005 Paris, France; clemence.basse@curie.fr (C.B.); romain.geiss@curie.fr (R.G.); pauline.durusquec@curie.fr (P.D.R.); catherine.daniel@curie.fr (C.D.); marieange.massiani@curie.fr (M.-A.M.); a-livartowski@unicancer.fr (A.L.); 2Paris Saclay Campus, University Versailles Saint Quentin, 78035 Versailles, France; 3Biostatistic Department, Hôpital Institut Curie, St Cloud, 75005 Paris, France; matthieu.carton@curie.fr; 4Data Department, Hôpital Institut Curie, St Cloud, 75005 Paris, France; maud.milder@curie.fr

**Keywords:** lung cancer, survival, Institut Curie, real-life data

## Abstract

**Simple Summary:**

The prognosis of metastatic lung cancer has been largely modified with targeted therapies and immunotherapies. Whether this survival benefit for subgroups of patients is beneficial to all patients treated for lung cancer has never been explored. The Institut Curie is an anti-cancer center in Paris, France, treating patients with lung cancer. Since the 2000s, Institut Curie has implemented dematerialized medical records, allowing for a large overview of the survival of patients with lung cancer according to treatment evolution. We found that the survival of patients with metastatic lung cancer has improved over the past 20 years, mostly in NSCLC (non-small cell lung cancer) but not in SCLC (small cell lung cancer). SCLC remains a disease with poor prognosis for which novel therapies are needed.

**Abstract:**

Over the past 20 years, several innovative therapies have been implemented in the treatment of lung cancer that have had reported survival benefits in clinical trials. Whether these improvements translate into the clinic setting has not been studied yet. We retrospectively analyzed all patients consecutively treated at Institute Curie for metastatic lung cancer. Diagnosis date was used to define three periods, based on the approvals of novel treatment strategies in the first-line setting, including targeted therapies in 2010 and immunotherapy in 2018. Endpoints included Overall survival (OS), survival rate of 2 years and 5 years, and a conditional survival rate of 2 years (if still alive at 6 months from treatment initiation). A total of 673 patients were identified for Period 1—2000 to 2009, 752 for Period 2—2010 to 2017, and 768 for Period 3—2018 to 2020. Median OS in the whole cohort was 11.1, 15.5, and 16.2 months, respectively. Median OS for patients with NSCLC or SCLC was 11.2, 17.2, and 18.2 months, or 10.9, 11.7, and 11.2 months, respectively. The two-year conditional survival was more favorable for NSCLC than SCLC patients. Outcomes were statistically higher for women as compared to men in all periods and all subgroups. Survival of patients with metastatic lung cancer has improved over the past 20 years, mostly in NSCLC, along with the implementation of novel treatment strategies.

## 1. Introduction

Lung cancer is the leading cause of cancer-related death, and the pathology is often diagnosed at metastatic stage [1]. Non-small cell lung cancer (NSCLC) and small cell lung cancer (SCLC) represent, respectively, 80–85% and 10–15% of the disease [2,3]. In the 2000s, median overall survival (OS) described in clinical trials was 7.9 months for advanced lung cancer patients treated with platinum-based chemotherapy in the first-line setting [4,5]. Over time, novel treatment agents were demonstrated to provide survival benefits for patients with lung cancer in the first-line setting. Novel treatment agents include targeted therapies in the setting of oncogene-addicted NSCLC, such as EGFR and ALK tyrosine kinase inhibitors, with access to these agents starting from early 2010 in France [6,7]. The median OS reported in clinical trials in the first-line setting with gefitinib (for *EGFR*-mutated adenocarcinoma) was 18.6 months [6], and with alectinib (for *ALK*-rearranged adenocarcinoma), it was not reached after close to 50 months of follow-up [7,8]. However, *EGFR*-mutated and *ALK*-rearranged lung cancers represent no more than 15% of patients with lung cancer. Novel treatment agents also include immunotherapy using anti-PD(L)1 immune checkpoint inhibitors with access to pembrolizumab in France, starting from early 2017 [9,10,11]. The median OS reported in the first-line setting with anti-PD(L)1, in association with chemotherapy, in NSCLC ranged from 22.0 months [12] to 17.1 months [13], respectively, for non-squamous and squamous histologies. Anti-PD(L)1 was available in France for SCLC in the first-line setting in association with chemotherapy in May 2020. The median OS reported in clinical trials in this population was 12–13 months under chemo-immunotherapy [11,14].

However, the clinical trials results were obtained from selected populations, and the translation into the clinical setting is not well explored. Patients in clinical trials are known to be fitter, younger, and with less comorbidities than routine care patients [15]. Whether and to what extent the implementation of these strategies, based on landmark randomized trials, translate into a survival improvement in a real-life setting is crucial for clinicians, patients, and global healthcare system to determine. There exists no study addressing this issue at a large scale of patients and over time. Moreover, whether and to what extent the improvement in the survival in subgroups of patients due to novel treatments contribute to the overall improvement in the survival for the whole cohort of patients with lung cancer has not been studied.

Here, we analyzed a large cohort of patients with metastatic lung cancer from a monocentric French anti-cancer center over the past 20 years from 2000 to 2020 to provide an insight into the survival outcomes based on the timeline of implementation of novel therapies.

## 2. Materials and Methods

### 2.1. Population

All consecutive patients diagnosed with primary metastatic lung cancer, initiating their treatment at Institute Curie from 1 January 2000 to 31 December 2020, were included. Patients with metachronous metastatic relapse were excluded from the cohort. Patients were considered metastatic if a first-line metastatic treatment was being administered in the first-line setting. Key variables included age, gender, and first-line treatment, including chemotherapy, immunotherapy (as single agent or in combination with chemotherapy), and targeted therapy.

### 2.2. Periods

We defined 3 periods based on the approvals in France of the new treatment strategies, in the first-line setting. Period 1 ranged from 2000 to 2009, when first-line treatment was only composed of platinum-based chemotherapy doublet [4,5]. Period 2 ranged from 2010 to 2016, after EGFR and ALK tyrosine kinase inhibitors were approved in patients with corresponding molecular alterations [6,7]. Period 3, between 2017 and 2020, corresponded to the era of immune checkpoint inhibitors as a single agent or combined with chemotherapy for both NSCLC and SCLC [9,10,11]. Diagnosis date was used to allocate patients in each Period.

### 2.3. Statistical Analysis

The cut-off date was 1st September 2022. Endpoints included Overall survival (OS), 2-year and 5-year survival rate, and conditional 2-year survival rate (defined as the overall survival rate if the patient were still alive at 6 months from treatment initiation). We used the Kaplan–Meier approach to obtain the survival curves, and the non-parametric Log-rank test was used to compare survival distributions.

## 3. Results

### 3.1. Population

A total of 2193 patients were included in the whole cohort. We included a total of 673 patients (31%) during Period 1, 752 patients (34%) during Period 2, and 768 patients (35%) during Period 3. There were 42%, 41%, 44% women for Periods 1, 2, and 3, respectively (Table 1). The median age at diagnosis was 61, 63, and 65 years, respectively. NSCLC was the major histology representing 81%, 83%, and 84% of patients for Periods 1, 2, and 3, respectively.

The median time between diagnosis and treatment initiation was similar across periods, with a median time of 1.3 months over the whole period of 2000–2020.

Across periods, there was a reduction in the use of first-line chemotherapy alone—from 96%, to 89%, and 55%, respectively, and an increased administration of targeted agents—from 2% to 7% and 15%, respectively, and immunotherapy (alone or in combination with chemotherapy)—from 0%, to 1% and 26%, respectively.

### 3.2. Evolution of Survival over the Three Periods

Median follow-up was 60.9 months (IC95%:56.1–69.1). Median OS was 11.1 months (IC95%:10.3–12.6), 15.5 months (IC95%: 13.0–17.7), and 16.2 months (IC95%:14.5–18.9) for Period 1, Period 2, and Period 3 (*p* < 0.0001). The 5-year survival rate increased over the three periods, with 12.4% [IC 10.1–15.4], 18.1% [15.4–21.2] and 18.5% [IC95%:14.7–23.2], as shown in Figure 1A, Table 2.

Patients with NSCLC experienced a significant increase in survival across the periods with a median OS of 11.2 months, 17.2 months, and 18.2 months, respectively (*p* < 0.0001), while this was not observed in SCLC for which the median OS was 10.9 months, 11.7 months, and 11.2 months, respectively, (Figure 1B,C). The 2- and 5-year survival rates for NSCLC were 26.4%, 11.9% and 39.6%, 19.5% and 43.4%, 20.1% for Period 1, Period 2 and Period 3, and for SCLC, 21.9%, 14.3%, and 24.8%, as well as 10.9, 27.1%, and 11.1%, respectively. The 2-year conditional survival rate was 36.7%, 51.0%, 57.6% for NSCLC, and 27.3%, 32.0%, 36.4% for SCLC, respectively.

### 3.3. Overall Survival

The median OS in the whole cohort was 13.9 months (IC95%: 12.9–15.5), Appendix A, Table 2.

The median OS for NSCLC was 14.9 months versus 11.2 months for SCLC (HR = 1.23, IC95% = 1.09–1.39, *p* = 0.001) (Appendix A). The 2- and 5-year survival rate was 36.9% and 17.4% for NSCLC versus 24.5% and 12.6% for SCLC (Table 2). The median OS for women was 17.6 months versus 11.9 months for men (HR = 1.33, IC95% = 1.21–1.47, *p* < 0.0001) (Appendix A, Table 2).

### 3.4. Gender Difference in Survival

Women had better a prognosis with a median OS of 14.2 months (IC95%: [11.8–16.4]), 20.2 months (IC95%: [16.9–24.2]), and 19.7 months (IC95%: [16.5–25.7]) for Period 1, Period 2, and Period 3, respectively, compared to men with a median OS of 9.7 months (IC95%: [8.6–11.1]), 12.4 months (IC95%: [11.0–15.5]), and 13.9 months (IC95%: [12.5–16.5]), respectively (Table 2).

Women had a better prognosis than men across all periods and that was consistent across histologies as well (Appendix A).

## 4. Discussion

Our study provides landmark survival data for more than 2000 consecutive patients with metastatic lung cancer grouped across three time periods over 20 years. We highlight the following results: (1) a doubling in the long-term survival of NSCLC patients between 2000 and 2020 (2-year survival rate of 25.5% for Period 1 compared to 40.8% for Period 3), suggesting that the results observed in clinical trials with novel treatment options can translate into clinical practice, (2) no significant survival improvement for SCLC, and (3) better outcomes in women across all periods, independent of histology.

The historical median OS was reported to be 6–8 months for NSCLC and 5 months for SCLC in the 2000s [4,16]. Our observed higher OS rates may be related to the improvement in patient selection for treatment (fit or unfit) over this period of time, with better imaging [17] and supporting care [18,19,20,21]. The OS improvements during Period 2 for NSCLC—with 2- and 5-year OS rates of 39.6% and 19.5%—suggest that targeted therapies, even if administered to 7.3% of patients in our cohort, could not only improve survival in this specific population, but have an effect on the long-term outcomes of the whole cohort, supporting the implementation of precision medicine approaches for patients eligible for such therapies [22]. Moreover, immunotherapy with PD(L)1 therapies as a first-line treatment may consolidate these survival outcomes, as reported in the literature from clinical trials [9,10]. Howlader et al. recently published mortality rates from lung cancer based on the SEER database. Concerning NSCLC, the mortality rate decreased annually from 2006 to 2013 by 3.2% and even faster from 2013 to 2016 by 6.3% annually [23], in line with our OS improvements. The advent of immunotherapy for treatment of NSCLC in France has not significantly improved survival in Period 3, 2018–2020, compared to the preceding Period 2, 2010–2017. This may reflect the short follow-up for patients in Period 3. An actualization of the results may be interesting in the years to come.

One strength of our study is highlighting the unmet needs of patients with metastatic SCLC, which are understudied. Even if immune checkpoint inhibitors combined with standard chemotherapy improve outcomes in clinical trials [9,10,11], the benefit remains modest without impact on OS in our cohort. This may be related to the heterogeneity of this population, with a significant proportion of patients not being eligible to immunotherapy. The 2-year survival rate, that better reflects long term outcomes and excludes early disease progression, improves over time as follows: 21.9%, 24.8%, and 27.1% in SCLC across periods without statistical difference. Howlader et al. also described a relative flat survival curve for SCLC for the 2001–2016 period [23]. In addition, the introduction of immunotherapy in SCLC started in 2020 in France, and the lack of long follow-ups for patients in our cohort who had started this novel combination therapy may impair the study’s ability to reflect the impact of survival on this subgroup of patients.

In our cohort, we observed a high proportion of women during Period 1 and Period 2 (42% and 41%, respectively, of the cohort), whereas in other French nationwide cohorts characterizing lung cancer incidence, the proportion of women during these Periods were closer to 30% or less [24]. This could be due to the recruitment of patients at Institut Curie, which is known to be specialized in cancers in women (such as breast and gynecologic cancers). Moreover, women have been described to develop lung cancer younger than men [25]. This can explain the global younger age at diagnosis of patients in our cohort compared to other series abroad [26].

Finally, we highlight the gender differences in the outcome of metastatic lung cancer. Women represented less than 15% to 30% of patients in landmark SCLC and NSCLC trials in Period 1 [4], close to 40% to 80%, respectively, in trials with targeted agents supporting strategies used in Period 2 [6,7,8], and from 35% to 50%, respectively, in trials with immunotherapy during Period 3 [9,10,11]. While the proportion of women with lung cancer is increasing worldwide [27,28], the outcomes in women were better, across the entire study period, independently from histology, which adds to previously reported evidence [21]. Several pieces of evidence support that women have a significant survival benefit after cancer diagnosis [29]. Smoking is more prevalent in males and is associated with cardiovascular comorbidities [30]. Biological differences are suspected to be associated with better cancer prognosis in women, such as gene expression, hormonal regulation, immune function, oxidative damage, and autophagy [31].

Further analysis on this topic could be of great interest for women, with more detailed data on molecular alteration and the type of treatment received.

This work has some limitations, such as the following: it does not describe in detail biomarker data, the type of treatment received, nor the stage at diagnosis and the sites of metastasis. These data were not available for patients in the whole cohort. However, the results in terms of survival show how novel therapeutics, as well as overall improvements in care, influence survival for a whole considered population. These are encouraging results for the development of new treatments for lung cancer.

## 5. Conclusions

To conclude, in this French monocentric cohort, the survival of patients with metastatic lung cancer significantly improved over the past 20 years in real-life data, along with the implementation of novel treatment strategies including targeted agents and immunotherapy. This is of major relevance for physicians, patients, and healthcare management for decision-making regarding approvals of new agents.

## Figures and Tables

**Figure 1 cancers-16-02768-f001:**
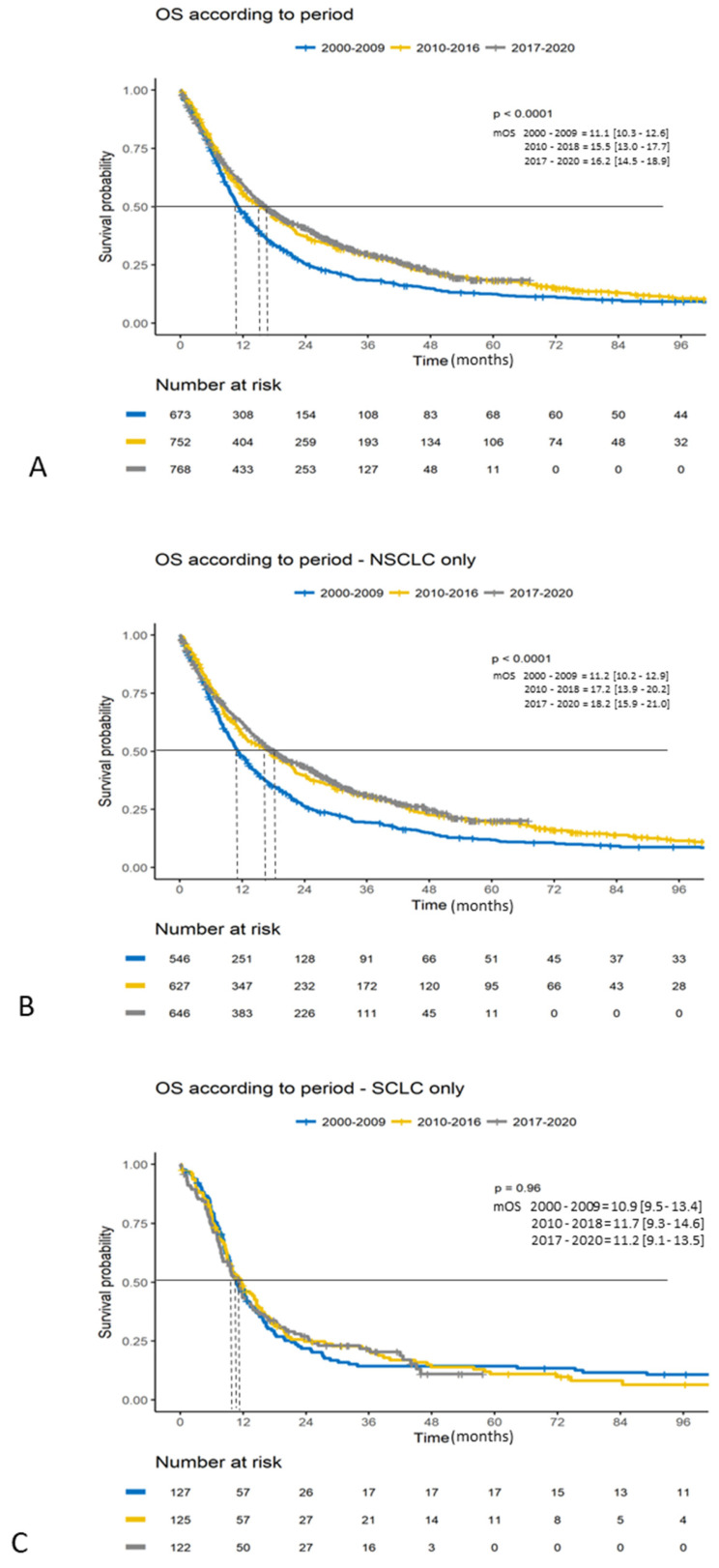
Overall survival curves according to Periods. (**A**) Per period. (**B**) Per period for patients with NSCLC. (**C**) Per period for patients with SCLC. Legend: NSCLC = Non-Small Cell Lung Cancer; SCLC = Small Cell Lung Cancer.

**Table 1 cancers-16-02768-t001:** Patients’ characteristics.

Characteristics	Period 1 -2000 to 2009-	Period 2 -2010 to 2017-	Period 3 -2018 to 2020-	Total
	*n* = 673 (100%)	*n* = 752 (100%)	*n* = 768 (100%)	*n* = 2193 (100%)
**Age at diagnosis**				
Median (years)	61	63	65	63
**Gender**				
Male (%)	394 (58)	442 (59)	430 (56)	1266 (58)
Female (%)	279 (42)	310 (41)	338 (44)	927 (42)
**Histology**				
NSCLC	546 (81)	627 (83)	646 (84)	1819 (83)
SCLC	127 (19)	125 (17)	122 (16)	374 (17)
**Time between diagnosis and treatment initiation**				
Median (months)	1.1	1.4	1.2	1.3
**Treatment administered**				
Chemotherapy alone	643 (96)	671 (89)	424 (55)	1738 (80)
Targeted therapy	14 (2)	55 (7)	114 (15)	183 (8)
Immunotherapy +/− CT	0 (0)	4 (1)	196 (26)	200 (9)
Other	16 (2)	22 (3)	34 (4)	72 (3)

Legend: CT = Chemotherapy; SCLC = Small Cell Lung Cancer; NSCLC = Non-Small Cell Lung Cancer; Other (for treatment administered): missing information or palliative care.

**Table 2 cancers-16-02768-t002:** Overall survival according to Periods.

	Period 1 2000–2009	Period 2 2010–2017	Period 3 2018–2020	*p*	Total Period 2000–2020	*p*
**Whole cohort**				<0.0001		-
mOS	11.1 [10.3–12.6]	15.5 [13.0–17.7]	16.2 [14.5–18.9]		13.9 [12.9–15.5]	
2-y surv	25.5 [22.4–29.2]	37.2 [33.8–40.9]	40.8 [37.3–44.5]		34.8 [32.8–36.9]	
5-y surv	12.4 [10.1–15.4]	18.1 [15.4–21.2]	18.5 [14.7–23.2]		16.6 [14.9–18.5]	
2-y cond.s	34.8 [30.7–39.4]	48.0 [44.0–52.3]	54.2 [50.1–58.6]		46.1 [43.7–48.7]	
**Histology**						=0.001
**NSCLC**						
mOS	11.2 [10.2–12.9]	17.2 [13.9–20.2]	18.2 [15.9–21.0]		14.9 [13.6–16.3]	
2-y surv	26.4 [22.8–30.5]	39.6 [35.9–43.7]	43.4 [39.6–47.6]	<0.0001	36.9 [34.7–39.3]	
5-y surv	11.9 [9.3–15.2]	19.5 [16.5–23.0]	20.1 [15.8–25.5]		17.4 [15.5–19.5]	
2-y cond.s	36.7 [32.1–42.0]	51.0 [46.7–55.8]	57.6 [53.2–62.3]		49.2 [46.5–52.0]	
**SCLC**						
mOS	10.9 [9.5–13.4]	11.7 [9.3–14.6]	11.2 [9.1–13.5]		11.2 [9.9–12.5]	
2-y surv	21.9 [15.6–30.6]	24.8 [18.0–34.1]	27.1 [20.0–36.6]	=0.96	24.5 [20.4–29.5]	
5-y surv	14.3 [9.2–22.2]	10.9 [6.4–18.8]	11.1 [7.8–18.3]		12.6 [9.3–17.0]	
2-y cond.s	27.3 [19.7–37.8]	32.0 [23.7–43.4]	36.4 [27.4–48.3]		31.7 [26.6–37.8]	
**Gender**						<0.0001
**Female**						
mOS	14.2 [11.8–16.4]	20.2 [16.9–24.2]	19.7 [16.5–25.7]		17.6 [15.8–20.0]	
2-y surv	33.7 [28.5–39.9]	44.3 [39.0–50.3]	46.1 [40.9–52.0]	<0.001	41.7 [38.6–45.1]	
5-y surv	15.9 [12.0–21.1]	22.4 [18.0–27.9]	20.8 [14.4–30.1]		20.3 [17.6–23.6]	
2-y cond.s	42.7 [36.5–49.9]	54.4 [48.4–61.0]	59.1 [53.2–65.7]		52.5 [48.9–56.3]	
**Male**						
mOS	9.7 [8.6–11.1]	12.4 [11.0–15.5]	13.9 [12.5–16.5]		11.9 [11.0–13.0]	
2-y surv	19.6 [16.0–24.2]	32.2 [28.0–36.9]	36.6 [32.2–41.6]	<0.001	29.7 [27.2–32.4]	
5-y surv	10.0 [7.2–13.7]	15.0 [11.9–19.0]	16.4 [12.2–22.1]		13.8 [11.8–16.2]	
2-y cond.s	28.3 [23.2–34.3]	43.0 [37.8–48.8]	50.1 [44.7–56.2]		41.0 [37.8–44.4]	

Legend: SCLC = Small Cell Lung Cancer; NSCLC = Non-Small Cell Lung Cancer; mOS = median overall survival (in months) with 95% Confidence Interval *[IC95%];* 2-y surv = 2-year survival rate (in %); 5-y surv = 5-year survival rate (in %); 2-y cond.s = 2-year conditional survival rate (in %) if patient still alive at 6 months after treatment intitiation.

## Data Availability

Patients’ data were collected in accordance with General Data Protection Regulations (GDPR) with validation of the study by Institut Curie ethics and data committees. These data can be available after a request to the PI (Professor Nicolas Girard). The absence of opposition from all patients to use their data was obtained before the study.

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
