# Peer review of "Real-World Survival Impact of New Treatment Strategies for Lung Cancer: A 2000–2020 French Cohort"

_cancers, 2024, doi:10.3390/cancers16152768_

Round 1

Reviewer 1 Report

Comments and Suggestions for Authors

The authors report a significant increase in real world survival of patients with NSCLC following the approval of EGFR/ALK-targeted therapies in France in the period 2010-2016.  The improved survival across the entire cohort during this period is highly unexpected given that only 7% of patients received targeted therapy (with 1% receiving Immunotherapy and 3% receiving other treatment).  The authors fail to show that improved survival is in fact related to treatment selection and not a consequence of improved overall care, including improved imaging and better supportive care.  It would be worthwhile investigating the actual survival of patients treated with targeted therapies or immunotherapies, as compared with those receiving chemotherapy over the same time frame.

It is noteworthy that the advent of immunotherapy for treatment of NSCLC in France has not yielded significantly improved survival over that of the immediately preceding period (2010-2016), despite comparatively high uptake of Immunotherapy.  This should be discussed, especially given the very significant cost of IT to the French taxpayer.

There is an error in Table 1.  In the period 2000-2010, 14 patients received targeted therapy, representing 2% of the study cohort, not 22%.

Author Response

Comment 1: The authors report a significant increase in real world survival of patients with NSCLC following the approval of EGFR/ALK-targeted therapies in France in the period 2010-2016.  The improved survival across the entire cohort during this period is highly unexpected given that only 7% of patients received targeted therapy (with 1% receiving Immunotherapy and 3% receiving other treatment).  The authors fail to show that improved survival is in fact related to treatment selection and not a consequence of improved overall care, including improved imaging and better supportive care.  It would be worthwhile investigating the actual survival of patients treated with targeted therapies or immunotherapies, as compared with those receiving chemotherapy over the same time frame.

Response1: Thank you for this comment. We added in the introduction part the reported survival in trials with the approval of EGFR/ALK-targeted therapies and anti-PD(L)1 therapy: In the 2000s, median overall survival (OS) described in clinical trials was 7.9 months for advanced lung cancer patients treated with platinum-based chemotherapy in the first-line setting. Over time, novel treatment agents demonstrated to provide survival benefits for patients with lung cancer in the first-line setting. Novel treatment agents include targeted therapies in the setting of oncogene-addicted NSCLC, such as EGFR and ALK tyrosine kinase inhibitors with access to these agents starting early 2010 in France. Median OS reported in clinical trials in the first line setting with gefitinib (for EGFR mutated adenocarcinoma) was 18.6 months, and with alectinib (for ALK rearranged adenocarcinoma) was not reached after close to 50 months of follow-up. However EGFR mutated and ALK rearranged lung cancer represent no more than 15% of patients with lung cancer.Novel treatment agents also include immunotherapy using anti-PD(L)1 immune checkpoint inhibitors with access to pembrolizumab in France starting early 2017. Median OS reported in the first line setting with anti PD(L)1 in association with chemotherapy in NSCLC ranged from 22.0 months to 17.1 months respectively for non-squamous and squamous histologies. Anti PD(L)1 were reimbursed in France for SCLC in the first line setting in association with chemotherapy in May 2020. Median OS reported in clinical trials in this population was 12-13months under chemo-immunotherapy.

We wrote at the beginning of the Discusison part:  Our higher observed OS rates may be related to improvement of patient selection over this period of time, with better imaging, and supporting care.

Comment2: It is noteworthy that the advent of immunotherapy for treatment of NSCLC in France has not yielded significantly improved survival over that of the immediately preceding period (2010-2016), despite comparatively high uptake of Immunotherapy.  This should be discussed, especially given the very significant cost of IT to the French taxpayer.

Response2: 

We thank reviewer#1 for this remark. We added in the discusison part : The advent of immunotherapy for treatment of NSCLC in France has not signaificantly improved survival in Period-3 2018-2020 compared to the precedent Period-2 2010-2017. This is probably due to the short follow-up for patients in Period-3. An actualization of the results may be interesting in the years to come.

Comment3: There is an error in Table 1.  In the period 2000-2010, 14 patients received targeted therapy, representing 2% of the study cohort, not 22%.

Response3: Thank you for this point. We corrected this error.

Reviewer 2 Report

Comments and Suggestions for Authors

The author examined the lung cancer survival using data from a single institution in France, and compare the survival 200-2009, 2010-2017 vs 2018 and 2020. The author concluded that the survival of metastatic lung cancer has been improving over the past 2 decades, mostly in NSCLC but not SCLC. I have a few comments:

1. line 39, add citation after NSCLC and SCLC represents 80-85% and 10-15% of the disease. 

2. Table 1. In your Table 1, the targeted therapy % was 22% instead of 2%. Please double check.

3. Were the increase in targeted therapy over time and decrease in chemotherapy over time statistically significant? Please provide p-value for the trend change. 

4. Line 163: "..... related to improvement of patient selection over this period of time..." I don't quite understand what does author mean by patient selection. Please elaborate

5. The median age of lung cancer diagnosis is much younger than it is reported in the US ~ 70 years old. Could that contribute? Please elaborate.

6. Line 165: ".... indicate that targeted therapies, even if administered to 7.3% of patient population...." I am not sure if this could be a valid conclusion. The number of people who received targeted therapy is quite small only 55. I think it would be interesting to see the KM curve just for the targeted therapy population, and to tease out if that is actually the main driver. 

7. line 191:".....higher incidence of mutated NSCLC in Asian women". The author did not provide race/ethnicity of this population. I am not sure if this is a fair statement for this analysis. 

Author Response

Commen1: 1. line 39, add citation after NSCLC and SCLC represents 80-85% and 10-15% of the disease. 

Response1: Thank you for this remark. We added the following reference: Thai AA, Solomon BJ, Sequist LV, Gainor JF, Heist RS. Lung cancer. Lancet. 2021 Aug 7;398(10299):535-554.

Comment2: 2. Table 1. In your Table 1, the targeted therapy % was 22% instead of 2%. Please double check.

Response2: Thank you for this point. We corrected this error.

Comment3: 3. Were the increase in targeted therapy over time and decrease in chemotherapy over time statistically significant? Please provide p-value for the trend change. 

Response3: Thank you for this interesting point. The evolution of treatments repartition among periods was not however significant.

Comment4: 4. Line 163: "..... related to improvement of patient selection over this period of time..." I don't quite understand what does author mean by patient selection. Please elaborate

Response4: We developped this point and added with a reference the following text : “Better patient selection for treatment (fit or unfit), in addition to supportive care improves patients survival”.

NSCLC Meta-Analyses Collaborative Group. Chemotherapy in Addition to Supportive Care Improves Survival in Advanced Non–Small-Cell Lung Cancer: A Systematic Review and Meta-Analysis of Individual Patient Data From 16 Randomized Controlled Trials. J Clin Oncol. 2008 Oct 1; 26(28): 4617–4625

Comment5: 5. The median age of lung cancer diagnosis is much younger than it is reported in the US ~ 70 years old. Could that contribute? Please elaborate.

Response5: 

Thank you for this relevant point. We added in the discussion part:

In our cohort we observed a high proportion of women during Period-1 and Period-2 (42% and 41% respectiveky of the cohort), whereas in other French nationwide cohort chatacterizing lung cancer incidence the proportion of women during these Periods were closer to 30% or less. This can be due to the recruitment of patients at Institut Curie which is known to be specialized in women cancer (such as berast and gynecologic cancers). Moreover, women have been dsecribed to developp lung cancer younger than men. This can explain the global younger age at diagnosis of patients in our cohort compared to other series abroad.

Comment6: 6. Line 165: ".... indicate that targeted therapies, even if administered to 7.3% of patient population...." I am not sure if this could be a valid conclusion. The number of people who received targeted therapy is quite small only 55. I think it would be interesting to see the KM curve just for the targeted therapy population, and to tease out if that is actually the main driver. 

Response6: Thank you for this interesting point. However given the low number of pateints in this subrgoup we do not think that this will bring significant result. We modified the text as follows “ : “OS improvements during Period-2 for NSCLC – with 2- and 5-year OS rates of 39.6% and 19.5% - can suggest thatthat targeted therapies, even if administered to 7.3% of patients in our cohort, not only improved survival in this specific population, but have an effect on long-term outcomes in the whole cohort,”

Comment7: 7. line 191:".....higher incidence of mutated NSCLC in Asian women". The author did not provide race/ethnicity of this population. I am not sure if this is a fair statement for this analysis. 

Response7: Thank you for this point. We do not have ethnicity information. We ajusted the next as follow : “Women represented less than 15% to 30% of patients in landmark SCLC and NSCLC trials in Period-1, close to 40% to 80%, respectively in trials with targeted agents supporting strategies used in Period-2), and from 35% to 50% respectively in trials with immunotherapy during Period 3.” And suppres the not relevant part.

Reviewer 3 Report

Comments and Suggestions for Authors

The authors analyzed the differences in survival rates of lung cancer patients between 2000 and 2020. The analysis concluded that survival of patients with metastatic lung cancer has improved over the past 20 years, mostly in NSCLC (non-small cell lung cancer), but not in SCLC (small cell lung cancer). However, as can be seen from Figure 1, the difference between Period 2 and Period 3 is not significant.

Even for the significant difference between Period 1 and Period 2 and Period 3 is related to the evolution of treatment methods, the authors did not make a more detailed analysis. The authors can start with the treatment methods and analyze the data in depth.

The academic and clinical significance of the article are poor, and it is difficult to arouse the interest of readers.

The proportion of patients in Treatment administered in Period 1 in Table 1 is wrong.

Author Response

Comment1: Even for the significant difference between Period 1 and Period 2 and Period 3 is related to the evolution of treatment methods, the authors did not make a more detailed analysis. The authors can start with the treatment methods and analyze the data in depth.

Response1: Thank you for this comment. We however already know survivals according to treatment methods thanks to datas in clinical trials. Our point was to look at global survivals intergating different treatment strategies and their effective use.

Comment2: The academic and clinical significance of the article are poor, and it is difficult to arouse the interest of readers.

Response2: We disagree with this point. Real-life data of survival over 20 years in a specific population of patientes (lung cancer) treated with lung cancer is an opportunity to confirm the increasing survival among patients with NSCLC, also linked with improvement in supportive care. It also highlights the survival benefit in women compared to men. This work could be the start for other studies on gender survival in the disease with multinational approach. And it also highlights the poor improvements in SCLC management.

Comment3: The proportion of patients in Treatment administered in Period 1 in Table 1 is wrong.

Response3: Thank you for this point. We corrected this error.

Round 2

Reviewer 1 Report

Comments and Suggestions for Authors

I appreciate the edits made to the manuscript to soften the conclusion that the improved survival observed in period 2 over period 1 are entirely attributable to the advent of targeted therapies, however, I feel that the revised language doesn't go quite far enough.  In the section acknowledging weaknesses in the study I think that they need to overtly state that the possibility that survival benefit observed in this period may have more to do with overall improvements in care rather than the specific benefit of targeted agents cannot be excluded by the data shown.

In a similar vein, the phrase "probably due to" on line 230, p7, Discussion, should be replaced with "may reflect", as, at present, it suggests biased interpretation of their data.

Author Response

Comment 1: In the section acknowledging weaknesses in the study I think that they need to overtly state that the possibility that survival benefit observed in this period may have more to do with overall improvements in care rather than the specific benefit of targeted agents cannot be excluded by the data shown.

Answer1 : We thank the reviewer for this remark. We modified the discussion (l.225-230): However the results in terms of survival show how novel therapeutics, as well as overall improvements in care, influence survival for a whole considered population.

Comment 2: In a similar vein, the phrase "probably due to" on line 230, p7, Discussion, should be replaced with "may reflect", as, at present, it suggests biased interpretation of their data.

Answer2: 

Thank you for this comment, we adapted the text as follows : l.185

This is probably due to may reflect the short follow-up for patients in Period-3. An actualization of the results may be interesting in the years to come.

Reviewer 2 Report

Comments and Suggestions for Authors

no further comments

Author Response

Comment :

No further comment.

Response:

We thank you for your reply

Reviewer 3 Report

Comments and Suggestions for Authors

It's great that the authors believe their results are significant! I reserve my opinion of the paper. No further suggestions.

Author Response

Comment :

It's great that the authors believe their results are significant! I reserve my opinion of the paper. No further suggestions.

Response:

We thank you for your reply